# Combining Interference Lithography and Two-Photon Lithography for Fabricating Large-Area Photonic Crystal Structures with Controlled Defects

**Hongsub Jee [1], Min-Joon Park [2] , Kiseok Jeon [2], Chaehwan Jeong [2] and Jaehyeong Lee [1],***

1   Department of Electrical and Computer Engineering, Sungkyunkwan University, Suwon 16419, Korea; hsjee@skku.edu
2   Smart Energy & Nano R&D Group, Korea Institute of Industrial Technology, Gwangju 61012, Korea; pk5659@kitech.re.kr (M.-J.P.); jks8287@kitech.re.kr (K.J.); chjeong@kitech.re.kr (C.J.)
*   Correspondence: jaehyeong@skku.edu

**Abstract:** Interference lithography is a promising method for fabricating large-area, defect-free three-dimensional photonic crystal structures which can be used for facilitating the realization of photonic devices with a fast processing time. Although they can be used in waveguides, resonators, and detectors, their repeated regular array patterns can only be used for limited applications. In this study, we demonstrate a method for fabricating large-area photonic crystal structures with controlled defects by combining interference lithography and two-photon lithography using a light-curable resin. By combining regular array structures and controlled patterns, monotonous but large-area regular structures can be obtained. Furthermore, the patterned structures have considerable potential for use in various applications, such as solar cells, sensors, photodetectors, micro-/nano-electronics, and cell growth.

**Keywords:** interference lithography; two-photon lithography; fabrication; photonic crystals; polymer

## 1. Introduction

Recently, nanoporous structures have attracted considerable attention because of their unique applications in physics and chemistry, as well as potential technological applications, such as sensing, catalysis, DNA translocations, and templates for nanostructure self-assembly [1–7]. Nanoporous materials with large internal surface areas have been more extensively employed as host materials in nanotechnology compared with other nanomaterials [8]. For example, the pore size and porosity of nanoporous structures can be controlled such that these structures show a novel sink effect for capturing molecules [2]. Interference lithography (IL) utilizes the interaction between two or more coherent light sources to form periodic structures at nanoscale. This technique is used to obtain nanostructure materials on the order of the wavelength of light for optical applications, such as photonic crystals (PhC) [9–17]; however, it has limited usage because of its simple repeated array of patterns. To expand the usage of it, we studied two-photon polymerization. Two-photon lithography is a fabrication technique which uses the two-photon absorption process. Two-photon absorption is a third-order nonlinear optical absorption process in which two photons are absorbed to reach a real excited state [18]. As the name suggests, the resins are polymerized by simultaneously absorbing two photons, usually in the infrared region. It became widely exploited for the development of new concepts [19,20], such as optical computers [21], two-photon fluorescent microscopy [22], two-photon laser scanning fluorescence microscopy [23] and vascular grafts [24]. The two-photon lithography (2PL) is basically the same as direct laser writing, but with a key difference. Because the two-photon absorption is corresponding to the square of the intensity, the resin polymerizing is only in the ellipsoidal focus, and by moving the sample stage to the x, y, z direction, three-dimensional structures can be fabricated. However, the absorption of the two photons is

much less compared to one photon excitation, which necessitates the high intensity needed for 2PL. Because of this, sometimes certain wavelengths with energy equal to half the energy of a single photon absorption band are favored [25]. In this study, we used IL to produce regular repeated PhC structures and subsequently used 2PL to create controlled defects on the PhC structures. This approach can overcome the limitations of the PhC structures, which have only periodic structures [26], and facilitate their use for various applications.

## 2. Materials and Methods

IL was performed using a continuous-wave Nd:YVO4 laser (Verdi V6, Coherent, Santa Clara, CA, USA) operating at 532 nm. As shown in Figure 1A with a schematic image, the 532 nm beam was split into three equal-energy beams that were combined at the substrate to crosslink the patterning layer, and the angle between the beams and the substrate was 23°. Figure 1B shows the schematic image of interference at the substrate for fabricating PhC structures. The three beams which came from the same source coincided at the patterning layer on the substrate and the exposed area and crosslinked to make PhC structures. After exposure to the three beams for fabricating PhC structures, the sample was exposed to 2PL setup immediately without post process, such as post-baking and development, as shown in Figure 2. The 2PL setup used the 800 nm mode-locked Ti:sapphire laser (Mira 900, Coherent, Santa Clara, CA, USA) pumped by solid state laser (Verdi-V10, Coherent, Santa Clara, USA) with 130 fs pulses and a 76 MHz repetition rate. By moving the scanning stage, 2D or 3D structures can be patterned without photomask. The sample for patterning was prepared using a slide glass as the substrate. On top of the glass, we spin-coated 2.5% EPON resin SU-8 (Miller-Stephen- son Chemical Co., Danbury, CT, USA) with a photoinitiator (PC-2506, Polyset Co., Mechanicville, NY, USA) to improve the adhesion with a thickness of 5 nm between the patterning layer and the glass substrate. Subsequently, we spin-coated a 35% SU-8 layer of 2 μm thickness as the patterning layer.

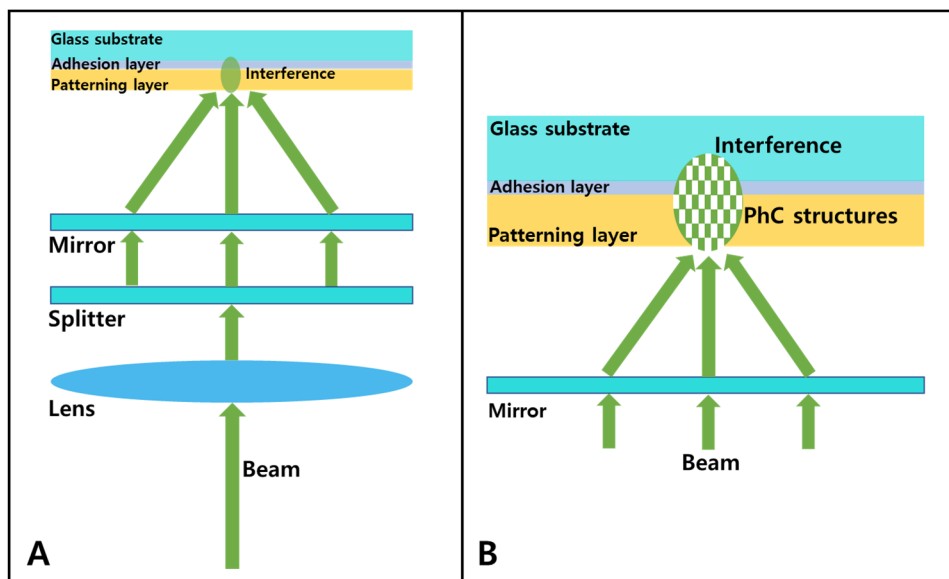

**Figure 1.** A schematic image of the IL setup (**A**) and interference at the substrate for fabricating PhC structures (**B**).

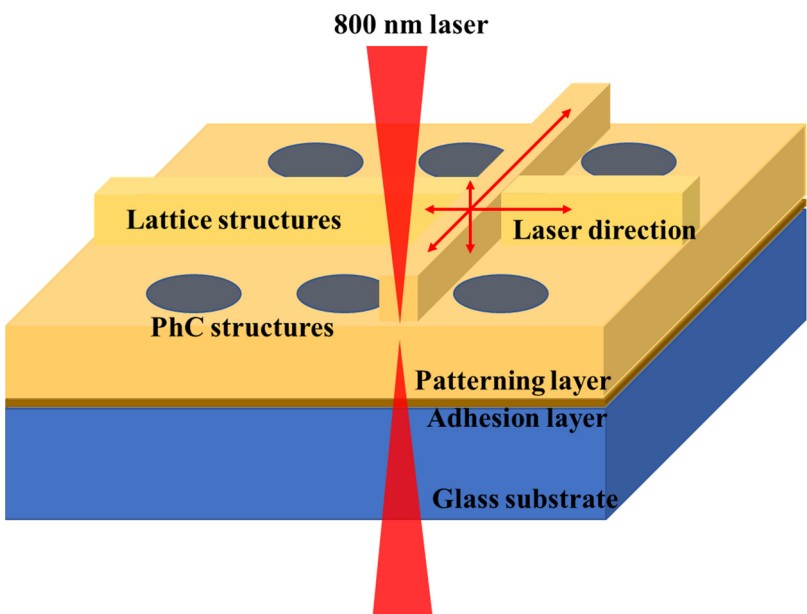

**Figure 2.** Second exposure with using two-photon lithography setup for patterning lattice structures by 800 nm laser and moving a computer-controlled piezoelectric stage.

After spin coating, the sample was prebaked at 65 °C for 30 min to evaporate the solvent and anneal out the stress caused by the spin coating. The 532 nm laser which has the size of 1 cm focused spot was used to irradiate with a cumulative dose of 0.6 W·cm$^{-2}$ for 3 to 8 s onto the sample to form a PhC pattern of area 1 cm$^2$. Then without post-baking, the 2PL system with an 800 nm femtosecond pulsed laser was employed using a microscope with an air objective lens (60×, NA 0.85 and 1.0, focused spot size < 1 μm) and computer-controlled piezoelectric stage to make a controlled defect to the PhC pattern. After exposure to the femtosecond pulsed laser, the sample was baked at 95 °C for 15 min to accelerate the acid-catalyzed reaction, which changes the solubility of the polymer and drives the diffusion of the photoactive compound [21]. The exposed sample was cooled sufficiently and then developed to complete the fabrication process.

## 3. Results and Discussion

In previous studies, we demonstrated and analyzed the characteristics of the PhC structures fabricated by IL [27,28]. The IL process produces PhC structures with regular repeated arrays, and Figure 3 shows the patterned structure. Because the computer-controlled 2PL process can fabricate designed patterns, we could fabricate lattice patterns with SU-8, as shown in Figure 4. To combine these two different structures, we performed IL followed by 2PL, and PhC structures with controlled defects were fabricated successfully. As mentioned earlier, we spin-coated 2.5% SU-8 on the glass substrate and cross-linked it first. This thin layer was applied as a buffer layer to increase the adhesion between the glass and the patterning layer, and then spin-coated the patterning layer. Initially, the IL and 2PL processes were performed without a buffer layer on the substrate, and Figure 5 shows the corresponding results. Without a buffer layer, the exposed patterning layer collapsed and shrank during the development process because of the weak adhesion force. SU-8 is hydrophilic and allows good film formation on general substrates via spin coating; however, the polarity changes and physical stress associated with thermal stress and shrinkage after crosslinking occured during the fabrication process, which may have caused the delamination of the film [29]. In addition, the resin used in this study consisted of monomers, oligomers, and photoinitiator molecules. When excited by highly localized near-infrared photons using a high-numerical-aperture objective lens with a femtosecond pulsed laser, monomers, oligomers, and photoinitiators are converted to solid polymeric forms with three-dimensional crosslinked networks via radical or cationic

polymerization [30–32]. As our approach uses 2PL for creating controlled defects on the PhC structures, determining the proper exposure dosage is important. Figure 6 shows the patterned structures with an insufficient exposure dosage. As shown, the lattice area patterned by the femtosecond pulsed laser was not exposed to a sufficient amount of energy; the lattice structures became curved, and the overall structures shrank. The following attenuation function was used to determine the exposure dosage in SU-8 resist $B(r)$:

$$B(r) = \alpha - \beta r + \gamma \times 10^{-6} \times r^2 \tag{1}$$

where $r$ is the depth of the resist from the surface (μm). The constants $\alpha$, $\beta$, and $\gamma$ are 0.7446, 0.0016, and 1.0378, respectively [30]. Based on this condition, scanning speed and the laser power varied from 0.005 to 0.05 μm/ms, and from 10 mW to 30 mW, respectively. To investigate the laser power-dependent correlations and the numerical aperture (NA) of objective lens, they were varied from 0.8 to 1.0 to find the optimized condition. The patterned structures with 0.05 μm/ms scanning speed was solid and wider than 0.005 μm/ms so the scanning speed was fixed at 0.05 μm/ms. Figure 7 shows the relationship between the laser power and linewidth under 0.05 μm/ms scanning speed. As the input power increased, the width of the patterned structures became wider, and the larger NA showed narrower linewidth with the same laser input power. Considering the several parameters for fabrication, we successfully fabricated PhC structures with controlled defects, as shown in Figure 8. Figure 8A shows the PhC structures, and Figure 8B shows the combined PhC and lattice structures. As Figure 8B shows, the fabricated lattice structures were patterned well on the PhC structures with good shape. As mentioned above, repeated PhC structures have fixed optical properties, and can only be used for limited applications. By adding controlled defects to the PhC structure, an optical state, for example, can be created within the forbidden bandgap frequency. As we performed IL first, followed by 2PL, areas patterned via 2PL were exposed twice, which caused morphological changes due to the high energy intensity, as shown in Figure 8C,D. The height of the lattice wall was 4 μm, whereas that of the PhC structure was 2 μm. The prepared sample had only a single patterning layer. Although there were two-step exposures, only once were post-baking and development performed. As the results show, the patterned structures have two different heights with different patterning. Accordingly, if these two-step fabrications can increase the height of the structure, we can reduce the overall cost and required processing time by properly designing the patterning for applications.

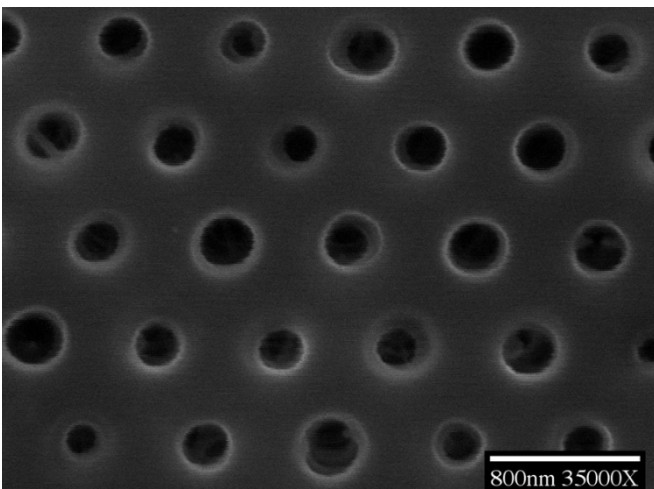

**Figure 3.** SEM image of the photonic crystal structures fabricated via interference lithography (top view).

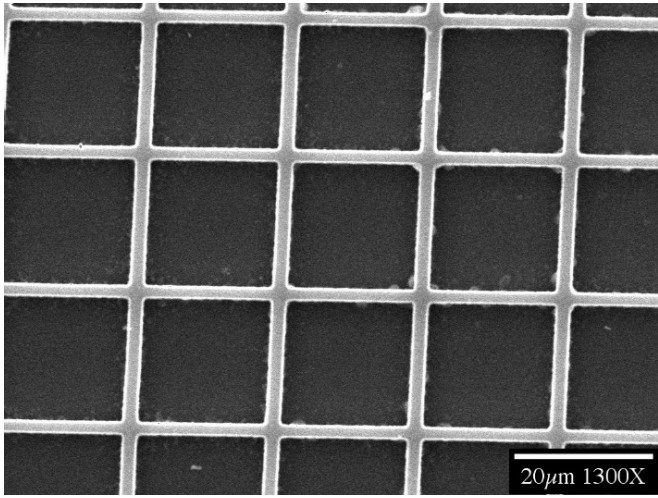

**Figure 4.** Lattice pattern fabricated via two-photon lithography (top view).

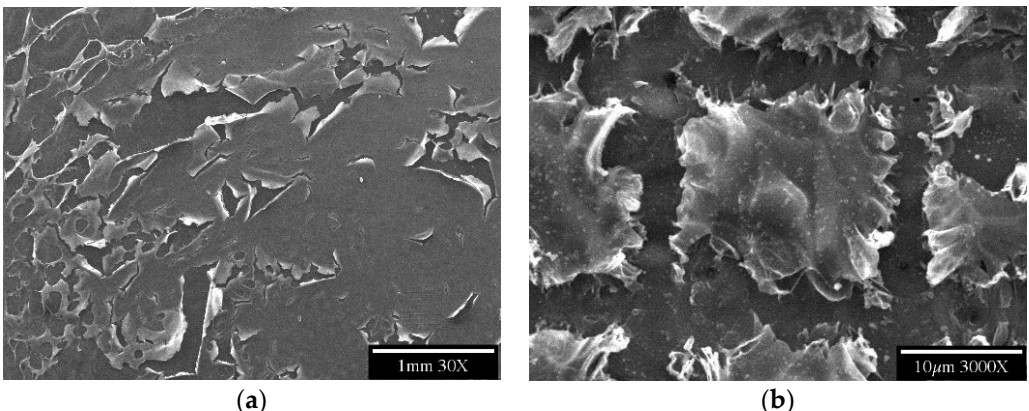

**Figure 5.** Delaminated IL patterned structures (**a**) and 2PL patterned structures (**b**) during the development.

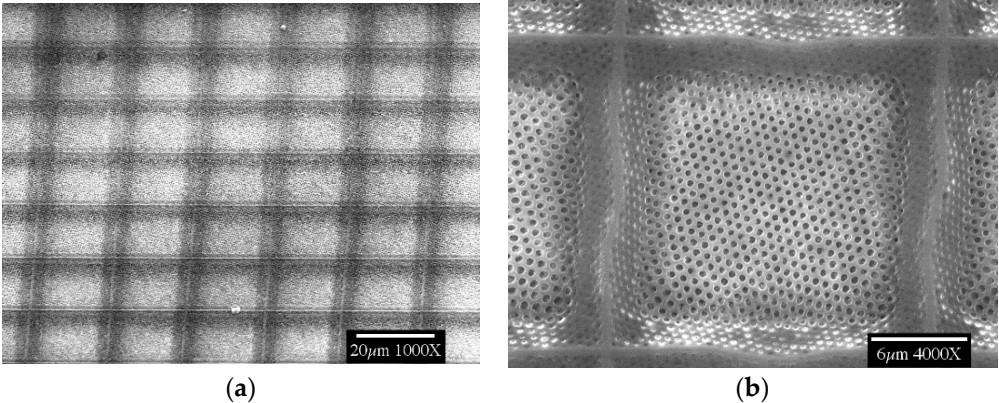

**Figure 6.** Collapsed patterned structures due to insufficient exposure dosage, 1000× magnification (**a**), 4000× magnification (**b**).

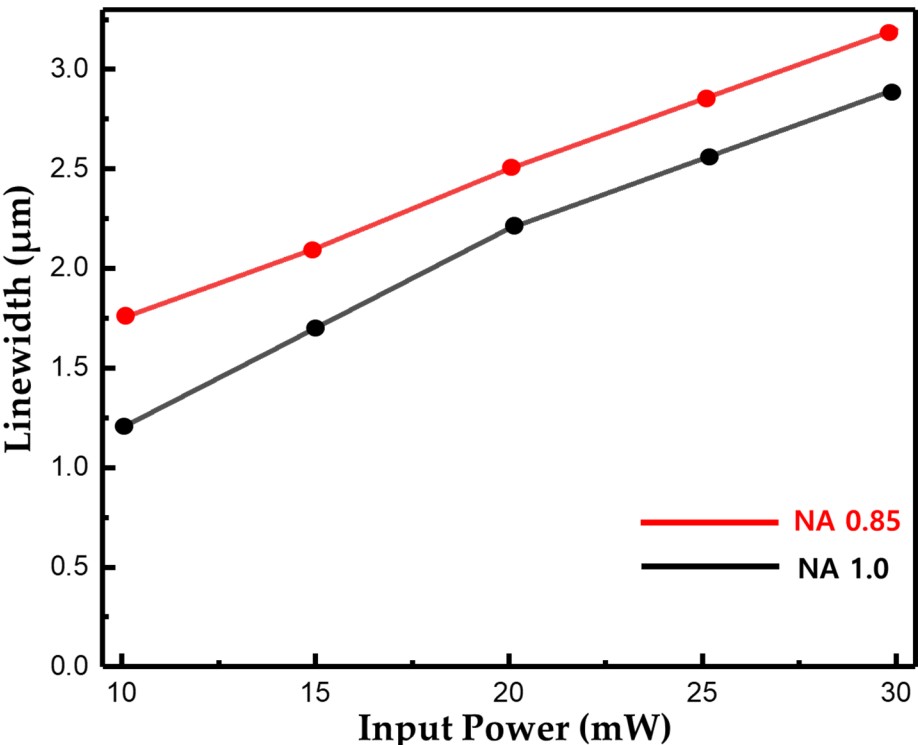

**Figure 7.** Relationship between the laser power and linewidth.

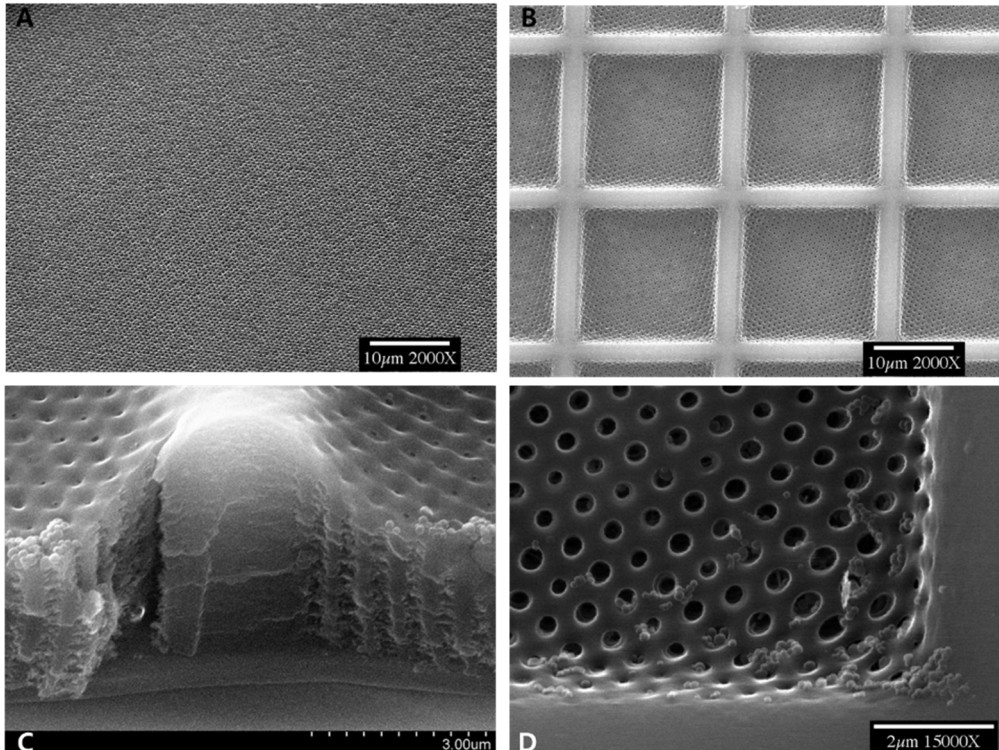

**Figure 8.** (**A**) PhC structures, (**B**) PhC structures with lattice pattern, (**C**) Double-exposed area, (**D**) Junction of the PhC and lattice pattern.

## 4. Conclusions

Two-step exposures with a single development were performed by combining IL and subsequent 2PL. We performed the IL first, followed by 2PL, to combine two different structures. While doing the 2PL, sample stage moved along the x, y, z direction and we

could fabricate lattice structures on top of the PhC structures successfully by optimizing the conditions. The 2PL process, using an 800 nm near-infrared laser, causes morphological changes to the double-exposed area owing to the high energy intensity. Consequently, we could fabricate a double-layered structure with single post-baking and development for patterning. Thus, this method can be used for fabricating double-layered structures with a simple, fast, and cost-effective fabrication method. Future studies are required to study the above phenomenon in depth and specific application will be suggested based on these results.

**Author Contributions:** Conceptualization, H.J., C.J., J.L.; methodology, H.J., M.-J.P.; investigation, H.J., K.J.; data curation, M.-J.P., K.J; writing—original draft preparation, H.J.; writing—review and editing, H.J., J.L.; supervision, J.L. All authors have read and agreed to the published version of the manuscript.

**Funding:** This study was supported by the Korea Electric Power Corporation (Grant number: R17XA05-1) and the National Research Foundation of Korea (NRF) grant funded by the Korea government (MSIT) (No. NRF-2019R1F1A1061615).

**Institutional Review Board Statement:** Not applicable.

**Informed Consent Statement:** Not applicable.

**Data Availability Statement:** The data that support the findings of this study are available from the corresponding author, J.L., upon reasonable request.

**Conflicts of Interest:** The authors declare no conflict of interest.

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
