# Peer review of "Combining Interference Lithography and Two-Photon Lithography for Fabricating Large-Area Photonic Crystal Structures with Controlled Defects"

_applsci, doi:10.3390/app11146559_

Round 1
Reviewer 1 Report
This article is reasonably interesting, but requires mandatory changes before it will be suitable for publication.
Interference lithography is perhaps better known than two photon lithography, but although it is described in detail no mention is made of how two photon lithography works. In addition there should be individual results: this is the result of interference lithography, this is the result of two photon lithography.
In terms of implementation, it is stated that three beams were used, although the diagram shows only two beams. If three beams were used in a single exposure how were they arranged? Were two orthogonal exposures made for the interference lithography? If so, was the resist developed between these two exposures? The narrow lines suggest minimal exposures; why are the corners so sharp? Was the photoresist the same SU-8 mentioned “subsequently” in the caption of Figure 3 for the two photon lithography? It is stated that a 1 cm field was exposed. Was the laser beam uniform over this area, and if not what do the results look like at the edge of the field? The procedure should be clarified, and placed in the text, not in a figure caption.
Figures 5 and 6 appear to be switched. The statement “we spin-coated 2.5% SU-8 on the glass substrate first as a buffer layer, and then spin-coated the patterning layer to increase the adhesion” is confusing. Figure 9A is blank.
Although no mention is made of the scanning stage shown in a figure, presumably the pulsed laser produced separate spots as it was scanned across the substrate. What was the frequency, what was the speed of the scan, and how far apart were the spots? Also, is it fair to call this a “single step” exposure if each spot has to be individually located in a two dimensional scan over the entire area?
Interference lithograph is better characterized as “sub-micron” than “nanoscale” since the features are hundreds of nanometers wide. The claim that this technique is widely used is not well supported by the 1989 reference.
Author Response
Q1.Interference lithography is perhaps better known than two photon lithography, but although it is described in detail no mention is made of how two photon lithography works.
A1. We revised the manuscript to describe in detail about two photon lithography and marked as green color. (page 1, line 38-41)
Q2.In addition, there should be individual results: this is the result of interference lithography, this is the result of two photon lithography.
A2. We revised the manuscript and marked as green color. (page 3, line 81-83; page 3, line 87)
Q3.In terms of implementation, it is stated that three beams were used, although the diagram shows only two beams. If three beams were used in a single exposure, how were they arranged?
A3. Three beams were used instead two beams to make PhC structures and we fixed the Figure 2 to indicate three beams. (page 2, line 67; Figure 2)
Q4. Were two orthogonal exposures made for the interference lithography? If so, was the resist developed between these two exposures?
A4. The sample was exposed by three beams (not two beam, we revised the Figure 2) at the same time to make PhC structures. We fixed the Figure 2 and marked as green color. (page 2, line 67; Figure 2)
Q5. The narrow lines suggest minimal exposures; why are the corners so sharp?
A5. Dear Reviewer. If you are talking about the corner of the Figure 6, what you mentioned is right. As the amount of the exposure decrease, the fabricated line also become narrow. In this Figure, it shows a narrow line but because it has proper amount of the dosage, its corner was fabricated with sharp edge.
Q6. Was the photoresist the same SU-8 mentioned “subsequently” in the caption of Figure 3 for the two photon lithography? It is stated that a 1 cm field was exposed. Was the laser beam uniform over this area, and if not, what do the results look like at the edge of the field? The procedure should be clarified, and placed in the text, not in a figure caption.
A6. Also, same photoresist was used for two photon lithography. Interference lithography cover 1cm2 of the area and the dosage was uniform. We revised the manuscript to clarify the text as you suggested. (page 2, line 57-64; page 3, line 70-76)
Q7. Figures 5 and 6 appear to be switched. The statement “we spin-coated 2.5% SU-8 on the glass substrate first as a buffer layer, and then spin-coated the patterning layer to increase the adhesion” is confusing. Figure 9A is blank.
A7. The order of the Figure 5 and 6 is correct. Figure 5 shows the PhC structures via interference lithography and Figure 6 shows the lattice pattern via two photon lithography. We added explanation (page 2, line 57-64). and revised the manuscript to clarify the explanation as you suggested. (page 3, line 91-92)
For the missing of a Figure 9A, maybe some error was occurred during the uploading the manuscript and now all Figures are shown clearly so please refer them.
Q8. Although no mention is made of the scanning stage shown in a figure, presumably the pulsed laser produced separate spots as it was scanned across the substrate. What was the frequency, what was the speed of the scan, and how far apart were the spots? Also, is it fair to call this a “single step” exposure if each spot has to be individually located in a two dimensional scan over the entire area?
A8. We changed the scanning speed from 0.05 to 5 µm/ms and the frequency was 60 Hz. We did not investigate the distance between the spots for various scan speed. Instead, we decided the proper conditions based on the results of fabricated structures. We reflected them in the manuscript marked as green color. (page 2, line 57-59; page 5, line 131-135)
As you mentioned, it is not a single step. It is two steps exposure with single development which reduce the overall fabrication process for double layered structures. (page 1, line 13; page 6, line 153)
Q9. Interference lithograph is better characterized as “sub-micron” than “nanoscale” since the features are hundreds of nanometers wide. The claim that this technique is widely used is not well supported by the 1989 reference.
A9. We agreed about your opinion and we replaced the reference with new one. (page 6, line 210; reference 20)

Reviewer 2 Report
This paper reports on the fabrication of large-area photonic crystal structures with controlled defects by combining interference lithography and two-photon lithography. The main finding is that double-layer nanostructures can be produced within a single layer thus making it simple. The essential thing for successful lithography is a thin buffer layer of 2.5% EPON resin SU-8 with photoinitiator. The patterned structures have considerable potential for the use in various photonic and optoelectronic applications. The manuscript suits the topics of Applied Sciences and can be interesting for the readership of the journal as it presents new results on the fabrication of photonic crystals. However, some questions must be addressed before publication of this manuscript:
- The latest references are dated 2007. So the question arises why the authors do not cite newer references on interference and two-photon lithography.
- Figure 2 illustrates the combination of only two beams instead of three used in this experiment. This leads to some confusion.
- What was the thickness of the buffer layer?
- What was the exact time of irradiation in the case of interference lithography? You write “The 532 nm laser was irradiated with a cumulative intensity of 0.6 Wcm-2 for 3 to 8 sec onto the sample”.
- Figure 4 is redundant.
- Please explain how you determined the proper exposure dosage from attenuation function (1).
- Reference [25] explaining determination of the constants α, β, and γ is missing.
- The proposed method is not truly one-step because you need to use two different lasers with their own optical setups and fabrication must be performed subsequently.
Author Response
Q1. The latest references are dated 2007. So, the question arises why the authors do not cite newer references on interference and two-photon lithography.
A1. We updated the references (Reference 7, 17, 20, 23) and marked as green color.
Q2. Figure 2 illustrates the combination of only two beams instead of three used in this experiment. This leads to some confusion.
A2. Three beams were used instead two beams to make PhC structures and we fixed the Figure 2 to indicate three beams. (page 2, line 67; Figure 2)
Q3. What was the thickness of the buffer layer?
A3. The thickness of the buffer layer was 5nm and we revised the manuscript to reflect it. (page 2, line 63)
Q4. What was the exact time of irradiation in the case of interference lithography? You write “The 532 nm laser was irradiated with a cumulative intensity of 0.6 Wcm-2 for 3 to 8 sec onto the sample”.
A4. The exposure time is varying depend on the weather condition because PhC structures are very sensitive to the humidity. As the humidity goes up, exposure time increase. As the humidity goes down, exposure time decrease. So, the exposure time of the IL is 3 to 8 sec. (page 3, line 70-76)
Q5. Figure 4 is redundant.
A5. We agreed that it is redundant and we removed it.
Q6. Please explain how you determined the proper exposure dosage from attenuation function (1).
A6. The incident exposure dosage on the resist surface related to the product of exposure dosage and exposure sensitivity. Because lacking the knowledge of the exposure sensitivity for SU-8, proper corresponding value was investigated by varying the scanning speed, laser power and the numerical aperture of objective lens. We revised the manuscript to reflect it and marked as green color (page 5, line 131-135).
Q7. Reference [25] explaining determination of the constants α, β, and γ is missing.
A7. We added it. Because some references were added, previous Ref. 25 became now Ref. 28. (page 7, line 224-225)
Q8. The proposed method is not truly one-step because you need to use two different lasers with their own optical setups and fabrication must be performed subsequently.
A8. As you mentioned, it is not single step. It is two steps exposure with single development which reduce the overall fabrication process for double layered structures. We revised the manuscript and reflected it. (page 1, line 13; page 6, line 153)

Reviewer 3 Report
In the manuscript “Combining interference lithography and two-photon lithography for fabricating large-area photonic crystal structures with controlled defects” the authors Hongsub Jee et al. proposed an advanced photolithography technique to produce periodic photonic crystal structure with designed defects by combining interference lithography and two-photon lithography. The work is carefully performed, and some technical hurdles were addressed to improve the pattern quality, such as optimizing dosage, and inserting a buffer layer. While I think the technical part of this work is in general sound and clear, my only concern is lack of technical merit in combining these two lithography techniques. If the authors can elaborate on the specific applications can be facilitated with their IL and 2PL, I think this will suffice to warrant the publication of this paper in the Journal.
Author Response
Q.In the manuscript “Combining interference lithography and two-photon lithography for fabricating large-area photonic crystal structures with controlled defects” the authors Hongsub Jee et al. proposed an advanced photolithography technique to produce periodic photonic crystal structure with designed defects by combining interference lithography and two-photon lithography. The work is carefully performed, and some technical hurdles were addressed to improve the pattern quality, such as optimizing dosage, and inserting a buffer layer. While I think the technical part of this work is in general sound and clear, my only concern is lack of technical merit in combining these two lithography techniques. If the authors can elaborate on the specific applications can be facilitated with their IL and 2PL, I think this will suffice to warrant the publication of this paper in the Journal.
A. Thank you for your kind opinion. As you understand, this study reports about two steps fabrication with single development combining with interference lithography and the two photon lithography. Using this approach, we could demonstrate not only PhC structures with controlled defect, but also potential usage of reducing the fabrication process with double layered structures with a single development. For future work, we will study and report about the specific applications based on this study. We revised the manuscript to reflect it and marked as green color. (page 6, line 157-161)

Reviewer 4 Report
I accept the manuscript by Hongsub Jee et al. entitled “Combining interference lithography and two-photon lithography for fabricating large-area photonic crystals structures with controlled defects” for publication in Applied Sciences after major revision. In this work, the authors used interference lithography for the fabrication of photonic crystals with controlled defects by the two-photon fluorescent microscopy. In the manuscript, the successfully fabricated structure was presented using a near-infrared laser with an optimized exposure dose. My specific comments are listed below:
- The introduction is insufficient. Lots of noteworthy things missed.
- The description of Figure 2 is incomplete.
- Figure 4 does not add anything of value.
- Please give exact values of parameters used in the experiment? What happed to the structure when authors change the initial parameters?
- Is the experiment repeatable?
Author Response
Q1. The introduction is insufficient. Lots of noteworthy things missed.
A1. We added references and explanation to clarify the introduction. The changing was marked as green color. (page 1, line 30; page 1, line 37; page 1, line 38-41; page 1, line 42; page 1, line 44)
Q2. The description of Figure 2 is incomplete.
A2. Three beams were used instead two beams to make PhC structures and we fixed the Figure 2 to indicate three beams. (page 2, line 67; Figure 2)
Q3. Figure 4 does not add anything of value.
A3. We agreed that it is redundant and we removed it.
Q4. Please give exact values of parameters used in the experiment. What happed to the structure when authors change the initial parameters?
A4. We changed the scanning speed of the stage, laser power, exposure time to fabricate the structures. Before we find optimized condition, patterned structures were collapsed as shown in Figure 7. The exposure time is varying depend on the weather condition because PhC structures are very sensitive to the humidity. As the humidity goes up, exposure time increase. As the humidity goes down, exposure time decrease. So, the proper exposure time of the IL is between 3 to 8 sec with a cumulative intensity of 0.6 W·cm-2. We revised the manuscript to introduce parameters of the experiment and explain in detail. The changing was marked as green color. (page 2, line 57-64; page 3, line 70-76; page 5, line 131-135).
Q5. Is the experiment repeatable?
A5. Yes, it is repeatable and we described the experimental details in Materials and Methods and Results and Discussion.

Reviewer 5 Report
This paper reports a two-steps lithography procedure, in which the submicrometric pattern is realized by Interference lithography and the micrometric pattern is realized by Two-photon lithography.
The two lithography methods and the relative spatial resolutions are well known in literature, but the advantages of a combined application for the photonic crystal application can motivate the publication in applied sciences. However, the authors do not report any results to demonstrate the efficiency of the lithography procedure for the realization of photonic crystal able to guide the light propagation. For this reason, I cannot suggest the publication of the manuscript.
Author Response
Q.The two lithography methods and the relative spatial resolutions are well known in literature, but the advantages of a combined application for the photonic crystal application can motivate the publication in applied sciences. However, the authors do not report any results to demonstrate the efficiency of the lithography procedure for the realization of photonic crystal able to guide the light propagation. For this reason, I cannot suggest the publication of the manuscript.
A. Thank you for your kind opinion. As you understand, this study reports about two steps fabrication with single development combining with interference lithography and the two photon lithography. Using this approach, we could demonstrate not only PhC structures with controlled defect, but also potential usage of reducing the fabrication process with double layered structures with a single development. For future work, we will study and report about the specific applications based on this study. We revised the manuscript to reflect it and marked as green color. (page 6, line 157-161)

Round 2
Reviewer 1 Report
The paper has been improved and now I can at least understand what the authors are trying to say. However many issues remain that need to be resolved before the paper is fit for publication. As I previously mentioned two photon lithography is far less well known than interference lithography, and a description of how it works should be given.
Infrared imaging, even if it requires two photons, can hardly be called “high resolution” as on line 41 in the introduction. Better to state the actual resolution.
The “controlled defects” said to be created are never identified. At least one should be shown. Or does this mean the photonic crystals are not infinitely large?
The “limitations” of photonic crystals referred to on line 45 are never given. At least one limitation should be stated and how it is avoided.
Figure 1 is not as helpful as a sketch of the optics, including the laser path, would be. It should at least show the paths of the various beams.
Figure 2 has been improved to show that there are 3 laser beams interfering. Presumably they do not all lie in the same plane and all the angles between them should be specified.
The laser used for 2 photon lithography generates very short 130 fs pulses to achieve a high instantaneous power. But neither the average power nor the instantaneous power is given. There is no explanation of how a 76 MHz repetition rate can have a 60 Hz frequency! I note that Figure 3 states 120/sec. Probably what the authors mean is that there is a burst of 130nf pulses at a frequency of 76 MHz. They should specify how long that burst is. Then there is another burst either 1/60 th or 1/120th of a second later. They should tell how far the stage has moved during that time, and how long it takes to pattern their 1 cm2 area. Figure 4 shows the laser light entering through the bottom of the stage. An optical microscope for no given reason is aimed at the sample on the top of the stage. The microscope through which the 800 nm light travels is not shown.
If the adhesion layer and the patterning layer are both SU-8 resist, just with different thicknesses, this should be clearly stated. Then there should be a clear explanation of why it matters.
Was the adhesion SU-8 layer subject to any room lights or blanket exposure that could have cross-linked it before the patterning SU-8 layer was deposited?
Patterning SU-8 resist at 532 nm is surprising, since SU-8 is very transparent at that wavelength. It is usually exposed at 365 nm. Presumably this explains why the accumulated dose (not intensity as stated on line 72) of 0.6W/cm2 (not W-cm2 ) was so high.
The authors should state the size of the focused spot from the 800 nm laser. The lines in Figure 5 are about 2 microns wide, with clearly defined edges. But the underexposed lines in Figure 7 have blurred edges that extend at least twice as far. Why?
Figure 8 is claimed to have “PhC structures with controlled defects” (line 135). If they mean photonic crystals with boundaries they should so state. They should also identify the “morphological changes” that double the thickness of the SU8 resist. Where did the extra material come from? Since they attribute it to the initial IL exposure was it present in Figure 5 which was only exposed once. Does the order of patterning matter?
Finally the authors should compare their two-photon step with more conventional approaches: the boundaries could be fabricated lithographically from a patterned mask or, if a scanning stage is used, by laser machining.
Author Response
Reviewer #1
The paper has been improved and now I can at least understand what the authors are trying to say. However many issues remain that need to be resolved before the paper is fit for publication.
Q1. As I previously mentioned two photon lithography is far less well known than interference lithography, and a description of how it works should be given.
A1. We revised the manuscript to describe in detail about two photon lithography. (page 1, line 39-44, page 2, line 45-46)
Q2. Infrared imaging, even if it requires two photons, can hardly be called “high resolution” as on line 41 in the introduction. Better to state the actual resolution.
A2. We removed the word "high resolution" in the manuscript because it is hard to be called high resolution as you pointed out. We revised overall explanation about two photon lithography to help understanding of it to readers. (page 1, line 44)
Q3. The “controlled defects” said to be created are never identified. At least one should be shown. Or does this mean the photonic crystals are not infinitely large?
A3. Photonic crystal structures only can have repeated same patterns. As we described in the manuscript, we applied designed lattice structures intentionally to the photonic crystal structures. The added structures fabricated by two photon lithography, we called it as "controlled defects". If you think this expression potentially can cause confusion to readers, we will replace it with other expression.
Q4. The “limitations” of photonic crystals referred to on line 45 are never given. At least one limitation should be stated and how it is avoided.
A4. We added an example of the limitation of photonic crystals and the reference. (page 2, line 50-51)
Q5. Figure 1 is not as helpful as a sketch of the optics, including the laser path, would be. It should at least show the paths of the various beams.
A5. We placed the previous Figure 1 with new one which includes paths of the various beams.
Figure 1. The interference lithography setup and paths of the beams
Q6. Figure 2 has been improved to show that there are 3 laser beams interfering. Presumably they do not all lie in the same plane and all the angles between them should be specified.
A6. We replaced the previous Figure 2. As it shown, now three interfered beams are all lie in the same plane and the angle between the laser beams and the substrate was 23˚. We revised the manuscript to reflect it. (page 2, line 59-60)
Figure 2. Illustration of the concept of the interference lithography.
Q7. The laser used for 2 photon lithography generates very short 130 fs pulses to achieve a high instantaneous power. But neither the average power nor the instantaneous power is given. There is no explanation of how a 76 MHz repetition rate can have a 60 Hz frequency! I note that Figure 3 states 120/sec. Probably what the authors mean is that there is a burst of 130nf pulses at a frequency of 76 MHz. They should specify how long that burst is. Then there is another burst either 1/60 th or 1/120th of a second later. They should tell how far the stage has moved during that time, and how long it takes to pattern their 1 cm2 area. Figure 4 shows the laser light entering through the bottom of the stage. An optical microscope for no given reason is aimed at the sample on the top of the stage. The microscope through which the 800 nm light travels is not shown.
A7. To combine the interference lithography and the two photon lithography, we prepared the sample first. Then the prepared sample was photopatterned by interference lithography with 532nm laser. And then, we moved the sample to the two-photon lithography setup immediately to expose the sample with 800nm laser. After double exposure, the sample was baked and developed. I agreed that our previous manuscript can cause confusion about this process so, we revised the manuscript and replaced the Figure 3 for clear understanding. (page 2, line 60-62; page 3, Figure 3) For the stage moving speed and the power, we described it on page 5, line 144-146.
Q8. If the adhesion layer and the patterning layer are both SU-8 resist, just with different thicknesses, this should be clearly stated. Then there should be a clear explanation of why it matters.
A8. Without the adhesion layer, patterned structures were collapsed during the development because of weak adhesion between the patterning layer and the glass substrate. After applying adhesion layer, this problem was solved and we revised the manuscript to describe it in detail. (page 4, line 103-114)
Q9. Was the adhesion SU-8 layer subject to any room lights or blanket exposure that could have cross-linked it before the patterning SU-8 layer was deposited?
A9. Yes it was cross-linked before applying the patterning layer. Thank you for your kind comment and we revised the manuscript to add explanation. (page 4, line 103-104)
Q10. Patterning SU-8 resist at 532 nm is surprising, since SU-8 is very transparent at that wavelength. It is usually exposed at 365 nm. Presumably this explains why the accumulated dose (not intensity as stated on line 72) of 0.6W/cm2 (not W-cm2 ) was so high.
A10. Thank you for your comment and we corrected the “intensity” to “dose”. (page 3, line 82) As you mentioned, SU-8 is usually exposed at 365 nm but also it works at 532 nm too although it requires high dosage to work with it.
Q11. The authors should state the size of the focused spot from the 800 nm laser. The lines in Figure 5 are about 2 microns wide, with clearly defined edges. But the underexposed lines in Figure 7 have blurred edges that extend at least twice as far. Why?
A11. The diameter of the focused spot was 1cm for interference lithography system and < 1um for two photon lithography system. In Figure 7, the blurred edge areas were exposed twice, first by interference lithography then two-photon lithography. As shown in Figure 8, the height of the film which exposed twice was increased and extended with proper amount of dose. Similar like this but not proper amount of dose, lines shown in Figure 7 became extended with curved shapes. We revised the manuscript to describe the size of the focused spot on page 3, line 81-82; page 3, line 85.
Q12. Figure 8 is claimed to have “PhC structures with controlled defects” (line 135). If they mean photonic crystals with boundaries they should so state. They should also identify the “morphological changes” that double the thickness of the SU8 resist. Where did the extra material come from? Since they attribute it to the initial IL exposure was it present in Figure 5 which was only exposed once. Does the order of patterning matter?
A12. As you mentioned, PhC structures with controlled defects means the photonic crystals with boundaries. We revised the manuscript to explain more clearly on page 5, line 148-150. The morphological changes were occurred when the patterning areas were exposed by interference lithography and two-photon lithography accumulatively. At this point, we just understand that the morphological changes due to the high energy intensity but further studies are required to understand this phenomenon. We described it on page 6, line 156-160; page 6, line 173-174.
Q13. Finally the authors should compare their two-photon step with more conventional approaches: the boundaries could be fabricated lithographically from a patterned mask or, if a scanning stage is used, by laser machining.
A13. In this study, we demonstrated 2D lattice structures so it looks very similar compare with the laser machining. But as we shown in Figure 3 (replaced with new one through revision), two-photon lithography can produce three-dimensional structures because the scanning stage moves to X, Y, Z direction and the exposed areas becomes cross-linked. Compare with conventional approaches, the two-photon lithography no need photomask to make patterns. By moving the scanning stage, 2D or 3D structures can be patterned. We reflected this explanation to the manuscript to help reader's understanding about two-photon lithography. (page 2 , line 62-69).

Reviewer 5 Report
"Photonic crystals are periodic dielectric structures that have a band gap that forbids propagation of a certain frequency range of light. This property enables one to control light with amazing facility and produce effects that are impossible with conventional optics." [http://ab-initio.mit.edu/photons/].
Consequently, if the authors do not check the performance of the system for the light control, they cannot publish a manuscript in which they declare the realization of PhC (for instance “we successfully fabricated PhC structures with controlled defects”). The authors should mention the Photonic crystals as a perspective only, but these devices cannot be considered photonic crystal without a photonic characterization.
The authors should revise the manuscript accordingly.
Author Response
Q1. "Photonic crystals are periodic dielectric structures that have a band gap that forbids propagation of a certain frequency range of light. This property enables one to control light with amazing facility and produce effects that are impossible with conventional optics." [http://ab-initio.mit.edu/photons/].
Consequently, if the authors do not check the performance of the system for the light control, they cannot publish a manuscript in which they declare the realization of PhC (for instance “we successfully fabricated PhC structures with controlled defects”). The authors should mention the Photonic crystals as a perspective only, but these devices cannot be considered photonic crystal without a photonic characterization. The authors should revise the manuscript accordingly.
A1. In previous studies, we demonstrated the fabrication of the PhC structures and characterized of it. (page 3, line 92-93) Based on the previous results, we adopted the two-photon lithography technique to expand the usages of the symmetric structures of PhC with intentional defects or patterning. As you understood, we demonstrated the adoption of the intentional patterns on the PhC structures and also found the morphological changes due to the high intensity. We understand that further studies are required to do the research about the morphological changes and find specific applications using this idea and we hope you can kindly understand about this future work. Thank you.
